# Effects of Cognitive Behavioral Group Program for Mental Health Promotion of University Students

**DOI:** 10.3390/ijerph17103500

**Published:** 2020-05-17

**Authors:** Soojung Lee, Eunjoo Lee

**Affiliations:** Department of Nursing, Kyungnam University, 7 Gyeongnamdaehak-ro, Masanhappo-gu, Changwon-si, Gyeongnam 51767, Korea; ecrystal@kyungnam.ac.kr

**Keywords:** nursing, cognitive behavioral therapy, depression, self concepts, interpersonal relations

## Abstract

This study aimed to explore the effects of a group cognitive behavioral program on depression, self-esteem, and interpersonal relations among undergraduate students. A non-equivalent control group pretest-posttest design was used. A convenient sample of 37 undergraduates (18 in the experimental group and 19 in the control group) at K university located in Changwon, South Korea was used. Data were collected from February 4, 2019 to June 18, 2019. The experimental group received eight sessions of the program, which were scheduled twice a week, with each session lasting 90 min. Collected data were analyzed using a chi-square test, Fisher’s exact test, independent *t*-test, and repeated measures ANOVA by SPSS/WIN 23.0 (SPSS, Inc., Chicago, IL, USA). The interaction of group and time was significant, indicating that the experimental group showed an improvement in depression, self-esteem, and personal relationship compared to the control group. A significant group by time interaction for depression, self-esteem, and personal relationship was also found between the two groups. The study results revealed that the group cognitive behavioral program was effective in reducing depression and improving self-esteem and interpersonal relation. Therefore, the group cognitive behavioral program can be used for promoting the mental health of students as well as for preventing depression in a university setting.

## 1. Introduction

The suicide rates in Korea increased by 9.5% year-on-year to 26.6 per 100,000 people as of 2018, the highest among the Organization for Economic Co-operation and Development (OECD) countries [1]. The causes of suicide may vary by age, but mental problems are common causes, with depression being an important factor directly related to suicide. Since recently, the number of university students suffering from depression has been increasing every year [2], and the nation’s incidence of major depressive disorder is the highest among those aged 20–29 years, which includes college students [3]. 

The post-adolescent college years are when students set their own goals with self-determination, and independence, and experience a significant shift in environment such as studying, personal relationships, employment, financial management, daily life management, and time management [4]. Adequate adaptations are required for the transition to college life, but some students face difficulties such as loneliness because of living alone, comparative consciousness with peers, academic and job stress, difficulties in interpersonal relationships, and economic independence [4]. Maladaptation may lead to negative symptoms and disorders such as amnesia, avoidance, stress, anxiety, and anger, whose likelihood of developing into an unsuitable aspect or serious depression in post-adult life, as well as into an impaired psychological and social development in college students, may increase [5].

Depression can get worse because college students have a high independence and autonomy and confide in peer groups first rather than seeking help from their parents and professors [6]. It is also highly likely that the need for professional help is greater, but students are passive in seeking help, and tend to be negative about psychotherapy, counseling, and psychiatric therapy [6].

The incidence and prevalence of mental health problems among undergraduate students are high, and although prevention and treatment are essential before any serious mental illness develops due to the large spillover effect, there is a lack of prevention and treatment for depression among college students compared to other age groups [2]. In the United States, college counseling centers, organized by the American College Counseling Association, are operated to prevent depression in college students, and programs such as cognitive behavior therapy, interpersonal relationships, computer training, individual feedback by e-mail, exercise, and stress training are conducted [2]. In Korea, there are only a few counseling centers specializing in suicide cases, mental health centers, and suicide prevention activities. Though there are counseling centers in universities, they only encourage individual counseling or suggest visiting doctors. It is urgent to develop mental health care programs applicable to college students in the current situation where the level of depression and the crime rate of Korean undergraduate students is high and which can prevent them from developing negative emotions such as depression.

In many studies and theories about depression, self-esteem serves as a risk factor for depression, and low self-esteem is known as a critical feature of depression [7]. The vulnerable model for depression also assumed that low self-esteem is a major factor causing depression [8]. Self-esteem also has a causal effect on the development and maintenance of depression, particularly through interpersonal and interpersonal channels [7]. That is, low self-esteem causes social avoidance, which hinders social support, associated with depression, and which reduces attachment and satisfaction in close relationships due to people becoming more negative about the behavior of those around them [9]. Because of the nature of the developmental stage, college students are most affected by their friendships and become more vulnerable to depression [6] because they depend on interpersonal relationships, such as professors and family members. Therefore, depression among college students is closely related to self-esteem and interpersonal relationships, so this can be seen as an important factor for preventing depression.

The cognitive behavioral program is a form of treatment that seeks to address behavioral and emotional problems by correcting negative cognition and that is based on the theory that in the course of cognitive, emotional, and behavioral interactions, individual behavior and emotions are determined by good cognitions [10]. This restructures one’s negative and dysfunctional cognition and that of other people to suit reality [11]; it changes emotions and behaviors, and has been shown to be effective in previous studies in reducing depression in university students [10,12,13]. Self-esteem and interpersonal relationships can also be described in relation to cognitive models. Low self-esteem causes interpersonal problems in relation to negative beliefs [14], and difficulties in interpersonal relationships in relation to key beliefs, assumptions, and negative automatic thinking [15], by recognizing information in a way that is biased toward one’s own distorted cognition [16]. Cognitive therapy helps to resolve interpersonal issues by deliberately reconstructing perceptions of these particular interpersonal styles [16]. Therefore, CBT can be expected not only to reduce depression but also to make positive changes or have positive effects on self-esteem and interpersonal relationships.

Looking at previous studies applying the cognitive behavioral program to college students, we identified the effects of variables such as assessment-absorbing perfectionist college students [12], attention deficit disorder propensity [17], positive changes in the perception of life stress, social support, suicidal thoughts [13], perceived stress, physical symptoms, and negative automatic thinking [18]. Previous studies that designed cognitive behavior programs mostly focused on problem behavior or maladjustment among undergraduate students, but there was no approach at a preventive level for mental health promotion based on depression, self-esteem, and interpersonal relationships.

On the other hand, a group cognitive behavioral program can compare one’s state with the state of others, and the more homogeneity one feels among one’s members, the more effectiveness one can expect in a psychological intervention in a common experience [13]. The study also said that in terms of cost-effectiveness, there is a better effect than with individual CBT [5].

Therefore, in this study, we aimed to conduct a group cognitive behavioral program focusing on cognitive processes and behavioral changes to improve the mental health of undergraduate students to identify how the factors of depression, self-esteem, and interpersonal relationships are changed through a pre-test and post-test. This was expected to reveal the usefulness of a collective program of cognitive behavior for the mental health of undergraduate students and to serve as a basis for nursing interventions to prevent depression.

## 2. Materials and Methods 

### 2.1. Setting and Sample

This is a quasi-experimental trial to identify the effects of developing a group cognitive behavioral program for mental health promotion in undergraduate students. Participants were recruited into a convenience sample through a recruitment advertisement at K university located in Changwon, South Korea, between 4 February 2019 and 18 June 2019. Koreans are more culturally concerned about stigma related to mental illness than foreigners, so it was difficult to recruit participants with a risk of depression. Participants also had to adjust their time for the group program and be able to express their thoughts within the group. Because of these reasons, participants who easily agreed to engage in the group program were first assigned to the experimental group in view of the participation time and grade, and the rest were assigned to the control group. 

The sample size for the participants was calculated using the G*Power 3.1.2 program. The minimum sample size required for a *t*-test with α = 0.05, power β = 80%, and effect siz 0.40, based on a previous study [18], was 36 subjects in both groups. Considering the dropout, we planned to recruit 20 participants in each group. Among them, two in the experimental group and one in the control group dropped out of the group. Overall, the study sample comprised 37 participants: 18 in the experimental group and 19 in the control group. 

All participants met the following inclusion criteria: (1) an undergraduate student; (2) having the ability to read, understand, and communicate; and (3) agreeing to participate voluntarily in this study. Exclusion criteria for the study were as follows: (1) serious medical illness; (2) severe depressive symptoms (hallucinations, delusions), and behavioral disorders. The study was approved by the institutional review board of the university (Approval no. 1040460-A-2018-064), and all students signed the informed consent form.

### 2.2. Procedure

In this study, strengthening depression, self-esteem, and interpersonal relationships and having a positive self-image among undergraduate students was the main focus of attention in the group cognitive behavioral program. The contents of the program were based on the literature [19] applying the theory of cognitive therapy, and the analysis of previous studies applying the cognitive behavioral model. Depression focused on the content and process of negative thinking and cognitive vulnerability [20], and self-esteem focused on identifying self-concepts, experiences of praise and reward, and experiences of achievement [14]. Interpersonal relationships focused on issues such as identifying beliefs and assumptions about oneself and others in interpersonal situations, and intimacy, assertions, relationships, and maintenance issues [16]. The content validity and applicability of the program were received from a psychiatric nursing professor and a counseling professor at the student counseling center.

The program was based on understanding the cognitive behavioral model, effective linkages between cognitive and therapeutic interventions, synchronizing program participation, and strengthening training. The elements of the theoretical framework and the interventions provided by the cognitive behavior group program are shown in Figure 1. 

The experimental group engaged in a cognitive behavioral group program twice a week for one month. The time of intervention per session was 2 h, with the total duration of the intervention being 16 h. The experimental group consisted of three groups, and one group consisted of 6–7 people. Because the recruitment of the subjects was difficult, the experimental participants did not receive an intervention at the same time. Because the intervention was conducted as soon as the number of groups was recruited, the duration of the data collection was extended.

The program consisted of eight sessions, and the details of each session are shown in Table 1. Participants assigned to the experimental group attended group sessions using a curriculum based on the new elements of the cognitive behavioral model to promote mental health. The contents of the group cognitive behavioral program included the following: (1) sharing their experiences that caused negative emotions and self-introduction, and setting goals to be achieved through this program; (2) distinguishing and understanding the process of cognitive–emotional behavior, which is the basic concept of cognitive behavior theory and describes an individual’s automatic thinking in the event or situation that caused the negative emotion; (3) learning the types of cognitive distortion, exploring their cognitive distortion through conversations with each other, and synchronizing to avoid this distortion; (4) cognitively reconstructing the individual’s automatic thinking in connection with the past session and talking about changes in emotion and behavior; (5) sharing their experiences in which cognitive distortion affected interpersonal relationships and activities, and planning new interpersonal relationships and activities; (6) talking about their experiences about the interpersonal relationship and activity strategies planned in the last session, and sharing with others experiences that influenced cognitive and emotional changes; (7) exploring their cognitive changes and sharing them with others; (8) sharing their experiences with changed thoughts and behaviors before and after participating in the program and the applicability in the future. The program allowed learners to participate through group activities, discussions, feedback, and assignments. In each session, education and various activities were conducted, and an active interaction within the group was achieved through discussions and feedback between group activities and participants. The task was to record the cognitive processes for events or situations that involved negative emotions during the week, and the researchers provided feedback on the cognitive and reconstruction processes described.

The pre-survey was measured one week before the program in both the experimental and control groups, and included questionnaires about general characteristics, depression, self-esteem, and interpersonal relationships. The time to complete the questionnaires was about 15–20 min. The post-survey was conducted immediately after the eighth session. The survey was conducted in a quiet university classroom. The experimental participants were provided meals and were given vouchers during the pre- and post-intervention. The control participants were provided with vouchers for the pretreatment and one-month follow-up assessments.

### 2.3. Measurements

#### 2.3.1. Depression

Depression was measured using the Beck Depression Inventory (BDI), designed by Beck et al. [10]. The BDI consists of 21 items, and each item is rated on a 4-point Likert scale (0 = least; 3 = most). Higher total scores indicate greater depressive severity, with total scores ranging from 0 to 63. Cronbach’s α was 0.94 in the previous study [21] and 0.81 in this study. 

#### 2.3.2. Self-Esteem

Self-esteem was measured using the Korean version of self-esteem, which was revised by Jon [22] based on the Self-esteem Scale developed by Rosenberg [23]. This 10 items tool, which uses a 4-point Likert scale, contains two dimensions: positive self (5 items) and negative self (5 items). Higher total scores indicate a higher self-esteem, with total scores ranging from 10 to 40. Cronbach’s α was 0.92 in the previous study [23] and 0.88 in this study. 

#### 2.3.3. Interpersonal Relationships

Interpersonal relationships were measured using the Relationship Chang Scale developed by Schlein [24], which was revised by Moon [25] and modified by Chun [26]. This 25-item tool, which uses a 5-points Likert scale, contains seven dimensions, including satisfaction, communication, trust, intimacy, sensitivity, openness, and understanding. Higher total scores indicate a higher interpersonal relationship, with total scores ranging from 25 to 125. Cronbach’s α was 0.88 in the previous study [24] and 0.79 in this study.

#### 2.3.4. General Characteristics

General characteristics included gender, religion, monthly allowance, smoking, drinking, and age.

### 2.4. Statistical Analyses

Statistical analyses were conducted using the SPSS/WIN version 23.0 program. Descriptive statistics were used to analyze the general characteristics and variables, including depression, self-esteem, and interpersonal relationships. The Chi-square test, Fisher’s exact test, and *t*-test were used to examine the homogeneity in the variables between the experimental and control groups. In order to verify the effect of the group CBT program on mental health by time between the experimental group and control group, a repeated measures ANOVA was performed. Two-tailed tests and a 5% significance level were used in all analyses. 

## 3. Results

### 3.1. Homogeneity Test for General Characteristics and Dependent Variables between Experimental and Control Groups

There were no differences between the two groups in terms of the general characteristics and study variables, including depression and self-esteem. However, there was a significant difference between the two groups in terms of interpersonal relationships (Table 2).

### 3.2. Effects of the Cognitive Behavioral Group Program

#### 3.2.1. Hypothesis 1

Depression has significant interactions between group and time (*F* = 12.48, *p* = 0.001) and time (*F* = 30.21, *p* < 0.001). When compared with the control, the depression in the experimental group showed a significant reduction (*t* = −6.48, *p* < 0.001) (Table 3).

#### 3.2.2. Hypothesis 2 

Self-esteem has significant interactions between group and time (*F* = 7.53, *p* = 0.010) and time (*F* = 8.74, *p* = 0.006). When compared with the control, self-esteem in the experimental group showed a significant reduction (*t* = 3.70, *p* = 0.002) (Table 3).

#### 3.2.3. Hypothesis 3

Personal relationship has significant interactions between group and time (*F* = 17.72, *p* < 0.001) and time (*F* = 29.59, *p* < 0.001). When compared with the control, personal relationship in the experimental group showed a significant reduction (*t* = 5.20, *p* < 0.001) (Table 3).

## 4. Discussion

This study attempted to examine the effect of a group program for mental health promotion on depression, self-esteem, and interpersonal relationships among undergraduate students. 

The BDI is an inventory measuring the attitudes and symptoms of depression and is divided into four alternative statements. The standard cutoffs are scores of 0–9 for the normal range, 10–15 for mild depression, 16–23 for moderate depression, and 24–63 for severe depression [10]. The mean score of the BDI before intervention in the study was 16.16 in the experimental group and 13.00 in the control group, which indicates more than mild depression in both groups. Participants may have been interested in depression because they saw the advertisement and applied it to the study. On the other hand, the mean score of depression in previous studies [5,12,27] applying cognitive behavioral therapy to undergraduate students was relatively higher than that in this study, ranging from 21.63 to 23.16. This is because the cutoffs were set to 21 [5], 18 [12], and 10 and more [27], respectively, and students who agreed to participate in the experiment by the therapist’s recommendation were included. The mean score of self-esteem before the intervention was 29.22 in the experimental group and 32.84 in the control group. In Park and Son’s study [28], the self-esteem score of female college students with a negative physical image was 24.50 in the experimental group and 25.13 in the control group, which was relatively lower than that in this study. Since female college students have a high correlation between physical satisfaction and self-esteem [28], self-esteem may be lower in female college students with a negative physical image. The mean score of interpersonal relationships before the intervention was 76.50 in the experimental group and 87.42 in the control group, which was significantly higher in the control group. Homogeneity between the two groups was not met because those with more willingness to treat depression were assigned to the experimental group. In other words, people with more problems in their personal relationships may feel more depressed and want some treatments. Hwang et al. [29] showed that the mean score of interpersonal relationships in undergraduate students was 84.68 in the smartphone addiction (SA) group and 85.78 in the normal group, which was relatively higher than that in this study. People with SA may be more positive for interpersonal relationships than depressed people because they have various relationships with people in virtual space.

The major findings of the study showed that the cognitive behavioral group program applied to undergraduate students significantly decreased depression scores, and significantly increased self-esteem and interpersonal relationships scores. This suggests that the cognitive behavioral group program is effective in improving the mental health of undergraduate students.

First, the cognitive behavioral group program for mental health promotion was effective in reducing depression among undergraduate students. In the previous studies [12,13,28], the reduction in depression in the group cognitive behavioral program was greater than that in the control group, which is consistent with the results of this study. Depression can be reduced because the cognitive behavioral program prevents negative thoughts from causing negative emotions by using automatic avoidance or suppression. On the other hand, a previous study [30] applying the cognitive behavioral program to disaster-affected children, showed that depression before and after intervention decreased in both the experimental and control groups, but that there was no significant difference between the two groups. The reason why the level of depression did not decrease significantly was that the treatment direction was focused on the traumatic event and the psychological distress it caused and was aimed at grades 4-6 of elementary school. In other words, it was more effective in reducing post-traumatic stress or anxiety than depression. Choi and Hyun’s study [31] also showed no effect on depression in the group cognitive behavioral program because it focused on recognizing the link between automatic thinking related to smartphone use and emotional, behavioral, and impulsive responses. The cognitive behavioral program commonly uses strategies to change thoughts to induce individual behavioral and emotional changes, but it is estimated that there is a difference in the reduction of depression depending on how the researcher reconstructs the cognitive behavioral program. In the process of cognitive reconstruction, finding and reconstructing the elements of cognitive distortion related to depression may reduce depression. In addition, negative emotions such as depression are minimized in the process of revealing and sharing individual negative emotions through peer groups. Therefore, screening high-risk students with depression and applying the cognitive behavior group program will be effective in reducing depressive symptoms and preventing depression.

Second, the group cognitive behavior program was effective in improving the self-esteem of undergraduate students. The cognitive behavioral program significantly improved the self-esteem of female college students with a negative physical image [28] and was effective in both positive and negative factors of self-esteem, even in chronic schizophrenia [32]. In a context similar to the results of this study, one can see that the deeper level of implicit self-esteem changed and that the association rate of self-positive adjectives was significantly improved through group cognitive behavioral therapy in patients with a social anxiety disorder [33]. Individuals with low self-esteem seek negative feedback from their neighbors to test their negative concepts [34], which further exacerbates them. The group cognitive behavioral programs help participants identify negative thoughts and beliefs about themselves, share them with others, receive feedback, and accept themselves with new beliefs, which are thought to improve self-esteem. However, Choi et al. [13] showed no significant change in self-esteem due to the short period of three days and the lack of practical implementation. Although the cognitive behavioral program was a short-term treatment, it was difficult to overcome the prejudice about information processing and cope with one’s own assumptions and beliefs in two and three days.

Third, the group cognitive behavioral program was effective in improving interpersonal relationships. The interpersonal relationship in the control group before the intervention was significantly higher than that in the experimental group. However, changes in personal relationships in the experimental group were higher than those in the control group. People who are inadequate with alternative relationships have limited and strict cognitive-interpersonal markers for others. They also anticipate the markers and act on them, which leads to interpersonal problems. The cognitive behavioral program helps modify them to suit reality and to identify behavioral characteristics associated with cognitive-interpersonal markers. However, Choi and Hyun’s study [31] conducted the cognitive behavioral program focusing on situations that cause interpersonal anxiety, automatic thoughts, and behavioral responses at that time. As a result, interpersonal anxiety did not decrease. The reason may be that the level of anxiety and depression in participants was not serious, and the focus of the study was on preventing smartphone overuse. This will allow subjects to modify their dysfunctional cognition and interpersonal markers, thereby enhancing interpersonal improvement. It is important to deal with the interpersonal relationships of undergraduate students because interpersonal relationships have a high correlation with psychiatric symptoms such as anxiety and depression and students experience problems in various relationships with friends, professors, and families. Attention to the high relationship between the interpersonal problems and mental problems of undergraduate students is necessary, and students with interpersonal problems should be provided with therapeutic interventions such as the cognitive behavioral program as well as expert counseling and interpersonal skills education.

## 5. Conclusions

In summary, the results of this study suggest that a group cognitive behavioral program for mental health promotion can reduce depression and improve self-esteem and interpersonal relationships among undergraduate students. Universities should screen students who are vulnerable to mental health and actively provide them with cognitive behavioral group programs. This administrative and financial support of the government and university is required to gather cognitive behavioral therapists possessing a wealth of experience. 

This study is meaningful in that it has led to a change in mental health by correcting cognitive distortions and changing irrational thinking through eight short-term cognitive behavioral group programs for the mental health promotion of undergraduate students. However, the limitations of this study are that there was no homogeneity between the experimental and control groups in the interpersonal relationships before the intervention and that the subjects were limited to students at only one university. In the future, cognitive behavioral group programs should be applied to a wider area and a large number of undergraduate students with high-risk depression, and long-term studies with designed randomized trials are needed to confirm the effects of this program.

## Figures and Tables

**Figure 1 ijerph-17-03500-f001:**
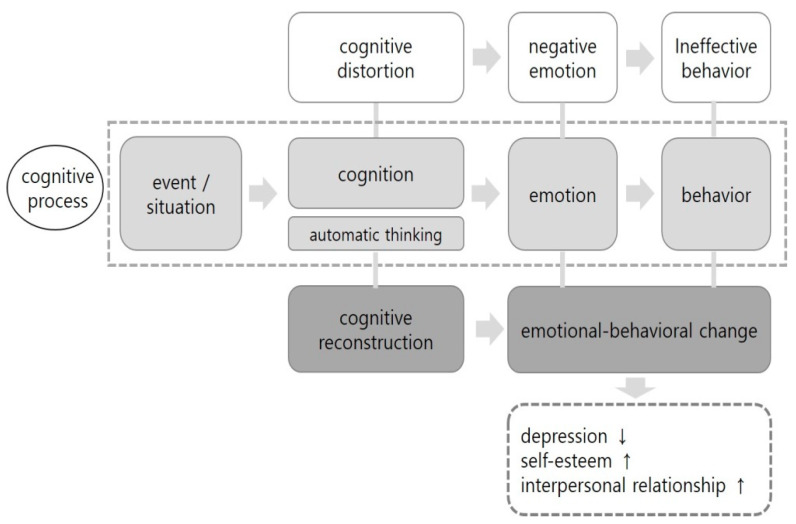
Conceptual framework of the cognitive behavioral group program.

**Table 1 ijerph-17-03500-t001:** The Contents of Group Cognitive Behavioral Program.

Session Topics		Contents	Activities
1	Understanding of experience and identifying individual characteristics	Orientation of program and pre-test. Sharing personal experience causing negative emotions.	small group individual exercise
2	Understanding the cognitive process	Understanding the process of cognition, emotion, and behavior. Description of an individual’s automatic thinking in an event/situation where one felt negative emotions.	Lecture small group individual exercise
3	Understanding the cognitive distortion	Understanding the type of cognitive distortions and identifying the cognitive distortions of the individual.	Lecture small group individual exercise
4	Reconstruction of cognitive process	Reconstructing the cognitive process and identifying changes in emotion and behavior.	Lecture small group individual exercise feedback
5	Planning of the interpersonal relationships and activities	Sharing the experience that cognitive distortion affected interpersonal relationships and activities, and planning the new interpersonal relationships and activities.	small group individual exercise feedback
6	Application of cognitive exercises	Sharing experiences in interpersonal relationships and activities, and identifying the effect on cognition and emotion.	small group individual exercise feedback
7	Identification of cognitive change	Explaining the change in an individual’s cognitive process.	small group individual exercise feedback sharing
8	Positive self-expression	Positive self-assessment and explaining the future applicability. Post-test.	small group feedback sharing

**Table 2 ijerph-17-03500-t002:** Homogeneity of the general characteristics and dependent variables between the experimental and control groups (*N* = 37).

Characteristics	Categories	Exp. (*n* = 18)	Cont. (*n* = 19)	χ2 or t	*p*
		M ± SD or *n* (%)	M ± SD or *n* (%)		
Gender ^†^	Male	2(11.1)	5(26.3)	1.39	0.238
	Female	16(88.9)	14(73.7)		
Religion ^†^	Christian	2(11.1)	4(21.1)	0.97	0.808
	Catholics	2(11.1)	1(5.3)		
	Buddhism	2(11.1)	2(10.5)		
	None	12(66.7)	12(63.2)		
Monthly allowance ^†^ (10,000 won)	20≤	2(11.1)	6(31.6)	3.26	0.353
	21–30	6(33.3)	6(31.6)		
	31–40	5(27.8)	2(10.5)		
	41≥	5(27.8)	5(26.3)		
Smoking ^†^	Yes	0(0.0)	3(15.8)	3.09	0.079
	No	18(100.0)	16(84.2)		
Drinking^†^	Never	6(33.3)	7(36.8)	3.60	0.165
	Once a month	11(61.1)	7(36.8)		
	More than once a week	1(5.6)	5(26.3)		
Age (yr) ^††^		22.44 ± 1.14	21.68 ± 1.56	1.67	0.103
Depression ^††^		16.16 ± 7.16	13.00 ± 6.53	1.40	0.168
Self-esteem ^††^		29.22 ± 5.87	32.84 ± 8.41	−1.50	0.140
Interpersonal relationship ^††^		76.50 ± 9.99	87.42 ± 10.85	−3.17	0.003

Exp. = Experimental group; Cont. = Control group; M = Mean; SD = Standard deviation; ^†^ Fisher’s exact test; ^††^ Independent *t*-test.

**Table 3 ijerph-17-03500-t003:** Effects of the cognitive behavioral group program on the dependent variables between the experimental and control groups (*N* = 37).

Variables	Group	Pre-Test	Post-Test	Source	*F*	*p*	Post-Pre	
		M ± SD	M ± SD				M ± SD	*t*(*p*)
Depression	Exp. (*n* = 18)	16.16 ± 7.16	6.72 ± 4.67	G	0.78	0.781	−9.44 ± 6.17	−6.48(<0.001)
	Cont. (*n* = 19)	13.00 ± 6.53	10.94 ± 7.50	T	30.21	<0.001	−2.05 ± 6.52	−1.37(0.187)
				G × T	12.48	0.001		
Self esteem	Exp. (*n* = 18)	29.22 ± 5.87	3.44 ± 5.79	G	0.46	0.500	4.22 ± 4.83	3.70(0.002)
	Cont. (*n* = 19)	32.84 ± 8.41	33.00 ± 8.90	T	8.74	0.006	0.15 ± 4.16	0.16(0.871)
				G × T	7.53	0.010		
Interpersonal Relation	Exp. (*n* = 18)	76.50 ± 9.99	89.72 ± 10.65	G	2.28	0.139	13.22 ± 10.78	5.20(<0.001)
	Cont. (*n* = 19)	87.42 ± 10.85	89.10 ± 12.83	T	29.59	<0.001	1.68 ± 5.01	1.46(0.160)
				G × T	17.72	<0.001		

Exp. = Experimental group; Cont. = Control group; M = Mean; SD = Standard deviation; G = Group; T=Time; G × T = Group × Time.

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
