# Peer review of "Effects of Cognitive Behavioral Group Program for Mental Health Promotion of University Students"

_ijerph, 2020, doi:10.3390/ijerph17103500_

Round 1

Reviewer 1 Report

This quasi experimental study is well described with clear articulation of background and significance and well linked with research hypotheses. Review of the literature is adequate. One recommendation would be to highlight that the study sample for from Korean university students and whether or not they were at the undergraduate and/or graduate level?

Also- it is unclear what the cultural implications are related to how mental health is constructed and conceptualized and what role stigma plays in expressions of depression, self esteem and interpersonal relationships. Although these were not the aims of this research it would be worth explaining the role of culture and how idioms of distress may be expressed in Korean youth.

The methods of pre and post test design are well described with appropriate test statistics used for the data. Unclear how the students where sampled, although there is a discussion of self selection. Was this a convenience sample?

Data collection methods are clearly described and reliability and validity of pre and post test measurement scales described. Analysis is provided in the discussion.

Conclusion and implications show statistical significant results as well as clinical significant recommendations for CBT group at low cost for students.

On first page second paragraph sentence beginning with 'Tailure to adapt will result in negative symptoms..." I recommend changing wording to may adapt to result in negative symptoms. Please clarify sample characteristics as limited to Korean university students.

Author Response

This quasi experimental study is well described with clear articulation of background and significance and well linked with research hypotheses. Review of the literature is adequate. One recommendation would be to highlight that the study sample for from Korean university students and whether or not they were at the undergraduate and/or graduate level?

→ They were university undergraduates.

Also- it is unclear what the cultural implications are related to how mental health is constructed and conceptualized and what role stigma plays in expressions of depression, self esteem and interpersonal relationships. Although these were not the aims of this research it would be worth explaining the role of culture and how idioms of distress may be expressed in Korean youth.

→ During the program, the researcher shared enough with the students about the role of culture and how idioms of distress.

The methods of pre and post test design are well described with appropriate test statistics used for the data. Unclear how the students where sampled, although there is a discussion of self selection. Was this a convenience sample?

→ Corrected to 'Subjects were recruited through a convenience sample'

Data collection methods are clearly described and reliability and validity of pre and post test measurement scales described. Analysis is provided in the discussion.

Conclusion and implications show statistical significant results as well as clinical significant recommendations for CBT group at low cost for students.

On first page second paragraph sentence beginning with 'failure to adapt will result in negative symptoms..." I recommend changing wording to may adapt to result in negative symptoms.

→ Corrected to 'Maladaptation may lead to negative symptoms'

Please clarify sample characteristics as limited to Korean university students.

Reviewer 2 Report

Major remarks

The authors describe the effects of an intervention in a very small sample of South Korean university students the aim of which is not clear. One of the keywords of the paper is „cognitive behavioral therapy”, and the authors use this term in the text several times, but CBT is delivered to patients, and there were no patients in this study. Mental health professionals and researchers should pay special attention to using proper technical terms and not refer to „therapy” when they do not work with patients.  

The sample selection, that is, the group of students who participated is another rather big problem. Selection was not based on risk of depression or bad mental health status or any kind of screening, pre-selection and matching, or at least randomization. Students signed up for participation, and those „who easily agreed to participate … were first assigned to the experimental group”. This means that the authors themselves introduced selection bias in their study. Judging from the depression scores of the participants, the experimental and control group did not significantly differ in terms of depression, and they were certainly not depressed. So then why deliver CBT-based intervention? Why to this group of students and not to another?

The study group was very small that resulted from an overestimation of effect size used for the calculation of sample size. This sample size does not allow any reasonable conclusion about the effectiveness of this program even if the authors’ conclusions would not be flawed.

Unfortunately, the authors’ conclusions are unfounded. The biggest problem shows up in the calculations of the Results. In 3.2.1, the authors misinterpret their results because they compare the post-pre DIFFERENCES in the case of all three outcome variables, rather than evaluating the changes comparing the 2 groups before and after.

Their sentences reflect a misunderstanding of what they have done. For example, depression mean score decreased BY 9.44 (not TO 9.44) in the experimental, and BY 2.05 (2.06 to be precise) in the control group.  The authors seemingly compare these two differences which does not convince the reader that depression scores were different between the two groups AFTER the program as they were not different before. The two groups’ post-test depression scores should have been compared as stated above. The same mistakes are committed for the statistical evaluation of self-esteem and interpersonal relation as well.

In light of the problems with the authors’ results, none of their major findings are supported by the data they presented.

Minor problems

Abstract

Author specification: what is Eunjoo Lee and PhD? Is PhD an author or the degree of Eunjoo Lee?

„K city” bears no information for the reader. If the authors do not want to name the city, they should at least tell some relevant information about it, eg. „a large city with one (two, three) university in South Korea”.

Statistical tests should be specified along with the variables for which the tests were used. Selection of the appropriate test depends on the variable. A simple list of tests does not suffice. The reviewer seriously doubts the appropriateness of the statistical tests because all statistical results were highly significant which virtually never happens in case of such small sample sizes. The intervention tested by the authors do not usually produce such dramatic results, not even in much bigger samples.

Introduction

The second sentence of the 3rd paragraph on page 2 is not clear: „The effect is well known in prior studies concerning a college student's cognitive and behavioral approach to depression…” ???

Methods

Why did the authors use an effect size of 0.40 for their sample size calculation? How did they calculate effect size?

If the program was 2x2 hours per week for 4 weeks, why did data collection take from February to June 2019? Was the intervention 90 minutes (Abstract) or 2 hours (p4)? Were all members of the experimental group receiving the intervention at the same time, in 1 group, or were they divided into several groups? On p4, this is written: „One group consisted of 6-7 participants” – Did the composition of the group(s)  change, as one would conclude from this sentence? If yes, why? One would think that interpersonal relationships are best built in stable groups.

2.3.4. „General characteristics” should be dissected into demographic variables (gender, age, income, religion) and behavior (smoking, drinking). The baseline features of the two groups should be presented along with statistical analysis regarding any difference. Since these are university students, their year of study and discipline should be taken into consideration. Different studies pose very different mental strains on students.

In 2.4, the authors list a number of parametric and non-parametric tests but nowhere in the manuscript do they describe whether their outcome variables were appropriate for parametric tests or if not, which variables were analysed by non-parametric test.

Results

Table 2. Christians and Catholics are separated into 2 groups – why? Catholics are Christians.

How could university students have monthly income?

As it is shown in Table 3, all 3 outcome variables improved both in the intervention and in the CONTROL group: depression scores decreased, self-esteem and interpersonal relation scores increased. This alludes to some positive external influence (confounding effect) or a secular, spontaneous change due to unidentified factors. These factors might have contributed to the decrease in the experimental group as well. The authors did not ponder this possibility and did not control for it.

The first sentences of all the hypotheses in the Results (3.2.1, 3.2.2., 3.2.3.) are not quite comprehensible. „The experimental group participating in the cognitive behavioral group program has lower depression scores than the control group was supported”-  do the authors mean that the hypothesis was supported? They did not actually prove it in any of the 3 hypotheses because they did not compare the post-test means of the outcome variables in the intervention and control groups.

Author Response

The authors describe the effects of an intervention in a very small sample of South Korean university students the aim of which is not clear. One of the keywords of the paper is „cognitive behavioral therapy”, and the authors use this term in the text several times, but CBT is delivered to patients, and there were no patients in this study. Mental health professionals and researchers should pay special attention to using proper technical terms and not refer to „therapy” when they do not work with patients.

→ It modified 'cognitive behavioral therapy' to 'cognitive behavioral program'.

The sample selection, that is, the group of students who participated is another rather big problem. Selection was not based on risk of depression or bad mental health status or any kind of screening, pre-selection and matching, or at least randomization. Students signed up for participation, and those „who easily agreed to participate … were first assigned to the experimental group”. This means that the authors themselves introduced selection bias in their study. Judging from the depression scores of the participants, the experimental and control group did not significantly differ in terms of depression, and they were certainly not depressed. So then why deliver CBT-based intervention? Why to this group of students and not to another?

→ The purpose of the study is to improve the mental health and prevent depression among university students. Selecting randomly assigned students with high depression scores may be a way to increase the reliability and validity of the study results. However, Korean college students had a stigma or fear of receiving psychological treatment, and it was difficult to select participants. For that reason, students who wished to receive CBT program were first assigned to the experimental group. CBT has usually been applied to patients, but it was intended to be applied as one of the non-pharmacological treatments to prevent students' depression based on previous studies. Therefore, the purpose of this study was not to treat depression or mental illness through CBT, but to identify changes in variables related to mental health in addition to depression.

The study group was very small that resulted from an overestimation of effect size used for the calculation of sample size. This sample size does not allow any reasonable conclusion about the effectiveness of this program even if the authors’ conclusions would not be flawed.

→ In this study, the effect size for measuring the number of samples is based on the results of a similar previous study, one of the methods for estimating the effect size.

Unfortunately, the authors’ conclusions are unfounded. The biggest problem shows up in the calculations of the Results. In 3.2.1, the authors misinterpret their results because they compare the post-pre DIFFERENCES in the case of all three outcome variables, rather than evaluating the changes comparing the 2 groups before and after.

Their sentences reflect a misunderstanding of what they have done. For example, depression mean score decreased BY 9.44 (not TO 9.44) in the experimental, and BY 2.05 (2.06 to be precise) in the control group. The authors seemingly compare these two differences which does not convince the reader that depression scores were different between the two groups AFTER the program as they were not different before. The two groups’ post-test depression scores should have been compared as stated above. The same mistakes are committed for the statistical evaluation of self-esteem and interpersonal relation as well.

In light of the problems with the authors’ results, none of their major findings are supported by the data they presented.

→ It was modified as follows

2.4. Statistical analyses

In order to verify the effect of the group CBT program on mental health by time between the experimental group and control group, repeated measures ANOVA was performed. Independent t-test was used to compare the difference between two groups for the amount of change of the main variables.

Table 3. Effects of Cognitive Behavioral Group Program on Dependent Variables between the Experimental and Control Groups (N=37).

Variables

Group

Pre-test

Post-test

Source

F

p

Post-pre

M±SD

M±SD

M±SD

t(p)

Depression

Exp. (n=17)

16.16±7.16

6.72±4.67

G

0.78

.781

-9.44±6.17

-3.53

(<.001)

Cont. (n=18)

13.00±6.53

10.94±7.50

T

30.21

<.001

-2.05±6.52

G*T

12.48

.001

Self esteem

Exp. (n=17)

29.22±5.87

3.44±5.79

G

0.46

.500

4.22±4.83

2.74

(.010)

Cont. (n=18)

32.84±8.41

33.00±8.90

T

8.74

.006

0.15±4.16

G*T

7.53

.010

Interpersonal

relation

Exp. (n=17)

76.50±9.99

89.72±10.65

G

2.28

.139

13.22±10.78

-3.45

(<.001)

Cont. (n=18)

87.42±10.85

89.10±12.83

T

29.59

<.001

1.68±5.01

G*T

17.72

<.001

 3.2.1. Hypothesis 1

The depression has significant interactions between group and time (F=12.48, p=.001) and time (F=30.21, p<.001). Depression decreased -9.44±6.17 points and -2.05±6.52 points respectively in the experimental and control group, and the difference in the amount of change was statistically significant (t=-3.53, p<.001) (Table 3).

 3.2.1. Hypothesis 2

The self esteem has significant interactions between group and time (F=7.53, p=.010) and time (F=8.74, p=.006). Self esteem increased 4.22±4.83 points and 0.15±4.16 points respectively in the experimental and control group, and the difference in the amount of change was statistically significant (t=2.74, p=.010) (Table 3).

 3.2.1. Hypothesis 3

The personal relationship has significant interactions between group and time (F=17.72, p<.001) and time (F=29.59, p<.001). Personal relationship increased 13.22±10.78 points and 1.68±5.01 points respectively in the experimental and control group, and the difference in the amount of change was statistically significant (t=-3.45, p<.001) (Table 3).

Minor problems

Abstract

Author specification: what is Eunjoo Lee and PhD? Is PhD an author or the degree of Eunjoo Lee?

→ It modified 'Eunjoo Lee, PhD2'

„K city” bears no information for the reader. If the authors do not want to name the city, they should at least tell some relevant information about it, eg. „a large city with one (two, three) university in South Korea”.

→ It modified 'at K university located in Changwon, South Korea'

Statistical tests should be specified along with the variables for which the tests were used. Selection of the appropriate test depends on the variable. A simple list of tests does not suffice. The reviewer seriously doubts the appropriateness of the statistical tests because all statistical results were highly significant which virtually never happens in case of such small sample sizes. The intervention tested by the authors do not usually produce such dramatic results, not even in much bigger samples.

→ Statistical analyses were conducted using the SPSS/WIN version 23.0 program. Descriptive statistics were used to analyze the general characteristics and variables, including depression, self-esteem, and interpersonal relationships. Chi-square test, Fisher’s exact test, and t-test were used to examine the homogeneity in the variables between experimental and control groups. Shapiro-Wilk test was used to test the normality of depression, self esteem, and interpersonal relationship between two groups, and variables that did not meet the normality assumption were analyzed using nonparametric methods. Independent t-test and Mann-Whitney U test were used to compare the difference between two groups for the amount of change of the main variables.

In order to verify the effect of the group CBT program on mental health by time between the experimental group and control group, repeated measures ANOVA was performed. Two-tailed tests and a 5% significance level were used in all analyses.

Introduction

The second sentence of the 3rd paragraph on page 2 is not clear: „The effect is well known in prior studies concerning a college student's cognitive and behavioral approach to depression…” ???

→ This restructures the negative and dysfunctional cognition of themselves and other people to suit reality [11], and changes emotions and behaviors, and has been shown to be effective in previous studies to reduce depression in university students [10,12,13].

Methods

Why did the authors use an effect size of 0.40 for their sample size calculation? How did they calculate effect size?

→ In this study, the effect size for measuring the number of samples is based on the results of a similar previous study, one of the methods for estimating the effect size.

If the program was 2x2 hours per week for 4 weeks, why did data collection take from February to June 2019? Was the intervention 90 minutes (Abstract) or 2 hours (p4)? Were all members of the experimental group receiving the intervention at the same time, in 1 group, or were they divided into several groups? On p4, this is written: „One group consisted of 6-7 participants” – Did the composition of the group(s) change, as one would conclude from this sentence? If yes, why? One would think that interpersonal relationships are best built in stable groups.

→ The time of intervention time was 2 hours. The experimental group consisted of 3 groups, and one group consisted of 6-7 people. Because the recruitment of the subjects was difficult, the experimental participants did not receive intervention at the same time. Because the intervention was conducted as soon as the number of groups was recruited, the duration of data collection was extended.

2.3.4. „General characteristics” should be dissected into demographic variables (gender, age, income, religion) and behavior (smoking, drinking). The baseline features of the two groups should be presented along with statistical analysis regarding any difference. Since these are university students, their year of study and discipline should be taken into consideration. Different studies pose very different mental strains on students.

→ We analyzed the differences according to the general characteristics of the two groups.

In 2.4, the authors list a number of parametric and non-parametric tests but nowhere in the manuscript do they describe whether their outcome variables were appropriate for parametric tests or if not, which variables were analysed by non-parametric test.

→ Shapiro-Wilk test was used to test the normality of depression, self esteem, and interpersonal relationship between two groups, and variables that did not meet the normality assumption were analyzed using nonparametric methods. Independent t-test and Mann-Whitney U test were used to compare the difference between two groups for the amount of change of the main variables.

Results

Table 2. Christians and Catholics are separated into 2 groups – why? Catholics are Christians.

→ Catholicism and Christianity are certainly different religions.

How could university students have monthly income?

→ monthly income → monthly allowance

As it is shown in Table 3, all 3 outcome variables improved both in the intervention and in the CONTROL group: depression scores decreased, self-esteem and interpersonal relation scores increased. This alludes to some positive external influence (confounding effect) or a secular, spontaneous change due to unidentified factors. These factors might have contributed to the decrease in the experimental group as well. The authors did not ponder this possibility and did not control for it.

→ We considered the confounding effect and any unidentitird factors. I tried to control it as much as I could, but now it seems difficult to control it.

The first sentences of all the hypotheses in the Results (3.2.1, 3.2.2., 3.2.3.) are not quite comprehensible. „The experimental group participating in the cognitive behavioral group program has lower depression scores than the control group was supported”- do the authors mean that the hypothesis was supported? They did not actually prove it in any of the 3 hypotheses because they did not compare the post-test means of the outcome variables in the intervention and control groups.

→ The results were verified through repeated measures ANOVA

Reviewer 3 Report

Thank you for the opportunity to review this manuscript.

The bones of the study are good, it's clear that a great deal of work went into the preparation of this paper. The article is well contextualized in the literature because the author/s explain the meaning of the different variables included in the research (depression, self-esteem, interpersonal relation and cognitive Behavioral therapy).

This study develops an interesting topic but, there are some items that need the author’s' attention, as follows:

With regard to the method, sample’s socio-economic context does not appear. Please, include this information and the university degree that participants are doing. I consider that the degree of difficulty of the university career can affect the results.

I propose to the authors that they extend the limitations of this study in terms of assessing the experimental group some time later to know if the program’s effect endure over time, not only using self-report measures, etc.

I ask the authors to clarify if there was any relationship between the students and the people who applied the program.

Thank you very much.

Author Response

The bones of the study are good, it's clear that a great deal of work went into the preparation of this paper. The article is well contextualized in the literature because the author/s explain the meaning of the different variables included in the research (depression, self-esteem, interpersonal relation and cognitive Behavioral therapy).

This study develops an interesting topic but, there are some items that need the author’s' attention, as follows:

With regard to the method, sample’s socio-economic context does not appear. Please, include this information and the university degree that participants are doing. I consider that the degree of difficulty of the university career can affect the results.

→ They were university undergraduates.

I propose to the authors that they extend the limitations of this study in terms of assessing the experimental group some time later to know if the program’s effect endure over time, not only using self-report measures, etc.

→ The results were verified through repeated measures ANOVA

I ask the authors to clarify if there was any relationship between the students and the people who applied the program.

→ There was no interest between the student and the researcher.

Round 2

Reviewer 2 Report

Major remarks

This is the second, revised version of an earlier manuscript which describes the effects of an intervention with no clear aim in a very small sample of South Korean university students.

The authors modified the manuscript where they could, for example removed some terms such as „therapy” that was not appropriate, and clarified the schedule of the program.

However, they obviously could not address other major problems that are rooted in the flawed design of the study. One such problem is sample selection which was severely biased. According to the authors: „People who easily agreed to participate in the group program were first assigned to the experimental group in consideration of the participation time and grade, and the rest were assigned to the control group.” This is a glaringly obvious selection bias that is absolutely critical, even more so because the sample size was so very small (18+19 persons altogether). Selection bias is an issue because an improperly selected sample does not allow the extrapolation of the conclusions to a larger population (such as university students in general) even if the findings are otherwise credible.

The authors’ conclusion that ”… group cognitive-behavioral program can be used for promoting mental health of students as well as preventing depression in a university setting” is simply untenable because of the selection bias.  

The authors still do not explain why they chose students with no mental problems, moreover, those with mental problems were excluded (NB: hallucinations and delusions are NOT depressive symptoms). Given the university setting, the reader simply does not understand why a quasi-experimental design was chosen instead of a before-after intervention when members of one sample receive the intervention, and pre-post scores of variables are compared.

The authors rephrased the results of their hypotheses but their statistical analysis is still not correct. Since the authors chose an experimental design, they should compare students in the experimental group to students in the control group before the intervention, and after. This is not what the authors did: instead, they compared the decrease in the experimental group to the decrease in the control group, and showed a significant difference. The reviewer took the data in Table3 and compared the BDI scores between the 2 groups before the intervention and after, and – as it could be suspected – there was no difference. In other words, the mean BDI score in the experimental group was NOT statistically different from the BDI score of the control group before the intervention, and the two mean scores were also not different after the intervention. Simply said, the intervention had no statistically significant effect in the experimental group compared to the control; the 2 groups were not different in terms of the outcomes before and after. The authors either lack basic comprehension of statistics to have carried out their analysis the way they did, or they knew exactly that the correct analysis will not yield significant results and chose an analysis that would.

In light of the flawed statistical analysis, the authors’ conclusion that ”… group cognitive-behavioral program can be used for promoting mental health of students as well as preventing depression in a university setting” is not even true for that 18 (17?) students who received the intervention.

The reviewer holds that the authors carried out a badly designed study and twisted the statistical analysis to receive statistically significant results. Their intervention did not significantly change either of the psychological variables of the students at the end of the intervention compared to before. All differences they found is due to chance and is not psychologically or interventionally meaningful.

The authors’ reliability and attention are further questioned by the changing numbers of the 2 groups: 18 in the experimental and 19 in the control group according to Chapter 2.1 and Table2, but 17 in the experimental and 18 in the control group in Table3. Moreover, these numbers were flipped in Table3 of the previous version of the manuscript.

In light of the basic problems with study design and flawed statistical analysis of the data, the reviewer does not recommend the publication of this study.

Author Response

This is the second, revised version of an earlier manuscript which describes the effects of an intervention with no clear aim in a very small sample of South Korean university students.

The authors modified the manuscript where they could, for example removed some terms such as „therapy” that was not appropriate, and clarified the schedule of the program.

However, they obviously could not address other major problems that are rooted in the flawed design of the study. One such problem is sample selection which was severely biased. According to the authors: „People who easily agreed to participate in the group program were first assigned to the experimental group in consideration of the participation time and grade, and the rest were assigned to the control group.” This is a glaringly obvious selection bias that is absolutely critical, even more so because the sample size was so very small (18+19 persons altogether). Selection bias is an issue because an improperly selected sample does not allow the extrapolation of the conclusions to a larger population (such as university students in general) even if the findings are otherwise credible.

The authors’ conclusion that ”… group cognitive-behavioral program can be used for promoting mental health of students as well as preventing depression in a university setting” is simply untenable because of the selection bias.

The authors still do not explain why they chose students with no mental problems, moreover, those with mental problems were excluded (NB: hallucinations and delusions are NOT depressive symptoms). Given the university setting, the reader simply does not understand why a quasi-experimental design was chosen instead of a before-after intervention when members of one sample receive the intervention, and pre-post scores of variables are compared.

→ Koreans are more culturally concerned about stigma related to mental illness than foreigners, so it was difficult to recruit participants with risk of depression. Participants had to adjust their time for group program and be able to express their thoughts within the group.

The authors rephrased the results of their hypotheses but their statistical analysis is still not correct. Since the authors chose an experimental design, they should compare students in the experimental group to students in the control group before the intervention, and after. This is not what the authors did: instead, they compared the decrease in the experimental group to the decrease in the control group, and showed a significant difference. The reviewer took the data in Table3 and compared the BDI scores between the 2 groups before the intervention and after, and – as it could be suspected – there was no difference. In other words, the mean BDI score in the experimental group was NOT statistically different from the BDI score of the control group before the intervention, and the two mean scores were also not different after the intervention. Simply said, the intervention had no statistically significant effect in the experimental group compared to the control; the 2 groups were not different in terms of the outcomes before and after. The authors either lack basic comprehension of statistics to have carried out their analysis the way they did, or they knew exactly that the correct analysis will not yield significant results and chose an analysis that would.

In light of the flawed statistical analysis, the authors’ conclusion that ”… group cognitive-behavioral program can be used for promoting mental health of students as well as preventing depression in a university setting” is not even true for that 18 (17?) students who received the intervention.

→ Table 3 has been revised. The main variables were compared to the differences between pre-and post-test in the experimental and control groups, respectively.

The reviewer holds that the authors carried out a badly designed study and twisted the statistical analysis to receive statistically significant results. Their intervention did not significantly change either of the psychological variables of the students at the end of the intervention compared to before. All differences they found is due to chance and is not psychologically or interventionally meaningful.

→ Unlike the control group, the experimental group had significant differences in depression, self-esteem, and interpersonal relation between pre- and post-intervention. (refer to Table 3)

The authors’ reliability and attention are further questioned by the changing numbers of the 2 groups: 18 in the experimental and 19 in the control group according to Chapter 2.1 and Table2, but 17 in the experimental and 18 in the control group in Table3. Moreover, these numbers were flipped in Table3 of the previous version of the manuscript.

→ Table 3 has been revised.

In light of the basic problems with study design and flawed statistical analysis of the data, the reviewer does not recommend the publication of this study.
